# Peripheral Blood Gene Expression Profiling Reveals Molecular Pathways Associated with Cervical Artery Dissection

**DOI:** 10.3390/ijms25105205

**Published:** 2024-05-10

**Authors:** Polina S. Shlapakova, Larisa A. Dobrynina, Ludmila A. Kalashnikova, Mariia V. Gubanova, Maria S. Danilova, Elena V. Gnedovskaya, Anastasia P. Grigorenko, Fedor E. Gusev, Andrey D. Manakhov, Evgeny I. Rogaev

**Affiliations:** 1Third Neurological Department, Research Center of Neurology, Moscow 125367, Russia; shlapakovaps@gmail.com (P.S.S.); kalashnikovancn@yandex.ru (L.A.K.); m.v.gubanova@yandex.ru (M.V.G.); gnedovskaya@mail.ru (E.V.G.); 2Department of Genomics and Human Genetics, Laboratory of Evolutionary Genomics, Vavilov Institute of General Genetics, Russian Academy of Sciences, Moscow 119333, Russiagusevfe@gmail.com (F.E.G.); 3Department of Genetics, Center for Genetics and Life Science, Sirius University of Science and Technology, Sochi 354340, Russia; manakhov@rogaevlab.ru (A.D.M.);; 4Center for Genetics and Genetic Technologies, Faculty of Biology, Lomonosov Moscow State University, Moscow 119192, Russia; 5Department of Psychiatry, UMass Chan Medical School, 222 Maple Ave, Reed-Rose-Gordon Building, Shrewsbury, MA 01545, USA

**Keywords:** cervical artery dissection, undifferentiated connective tissue dysplasia, genome-wide association studies, bulk RNA sequencing, nucleolar stress, senescence-associated secretory phenotype, mitochondrial malfunction, epithelial–mesenchymal plasticity

## Abstract

Cervical artery dissection (CeAD) is the primary cause of ischemic stroke in young adults. Monogenic heritable connective tissue diseases account for fewer than 5% of cases of CeAD. The remaining sporadic cases have known risk factors. The clinical, radiological, and histological characteristics of systemic vasculopathy and undifferentiated connective tissue dysplasia are present in up to 70% of individuals with sporadic CeAD. Genome-wide association studies identified CeAD-associated genetic variants in the non-coding genomic regions that may impact the gene transcription and RNA processing. However, global gene expression profile analysis has not yet been carried out for CeAD patients. We conducted bulk RNA sequencing and differential gene expression analysis to investigate the expression profile of protein-coding genes in the peripheral blood of 19 CeAD patients and 18 healthy volunteers. This was followed by functional annotation, heatmap clustering, reports on gene–disease associations and protein–protein interactions, as well as gene set enrichment analysis. We found potential correlations between CeAD and the dysregulation of genes linked to nucleolar stress, senescence-associated secretory phenotype, mitochondrial malfunction, and epithelial–mesenchymal plasticity.

## 1. Introduction

Cervical artery dissection (CeAD) is the primary cause of ischemic stroke in adults under 55 years old (accounting for up to 25% of all cases) [1,2,3]. CeAD occurs when the intima tears, leading to the formation of a double-lumen sign, intimal flapping, intramural hematoma with luminal stenosis, or sub-adventitial pseudoaneurysm [4,5]. More than 60% of CeAD cases occur in the internal carotid artery (ICA), 25% in the vertebral artery (VA), and the remaining 15% in a few cervical arteries simultaneously [6,7]. Up to 25% of CeAD cases are complicated by recurrent dissection within the first month [6]. Dissections typically heal within the first 3–6 months, so it is recommended to use anticoagulant or antiplatelet therapy during this period [2,8].

Despite advances in diagnostic imaging [4], the true prevalence of this disease remains underestimated due to its variable clinical severity, ranging from asymptomatic cases to fatal ischemic strokes [5]. Less than 5% of CeAD cases are linked to monogenic heritable disorders of the connective tissue (e.g., Ehlers–Danlos syndrome, Marfan syndrome, Loeyz–Dietz syndrome, osteogenesis imperfecta, and fibromuscular dysplasia) [9,10]. The remaining cases occur sporadically with established risk factors. Infection in the preceding 2–3 weeks, contraceptive use, pregnancy, arterial hypertension, hyperhomocysteinemia, and migraine increase the risk of CeAD, while hypercholesterolemia, diabetes, and obesity decrease the risk [2,9,11].

Patients with sporadic CeAD exhibit clinical signs of undifferentiated connective tissue dysplasia (UCTD), such as joint hypermobility, increased skin elasticity, elongation of the fingers, intervertebral disc hernias, mitral valve prolapse, and aortic root dilatation, more often than the general population [9,10]. Electron microscopy studies of skin biopsies have shown that 55–68% of CeAD patients have structural abnormalities in the extracellular matrix [12,13].

Pathomorphological studies in the area of vascular dissection have identified the following [6,14,15]:1.Disorganized thinning and straightening of the internal elastic membrane.2.Heterogeneous thickening of the tunica media with areas of non-inflammatory mucoid degeneration, necrosis, fibrosis, and calcification.3.Predominance of smooth-muscle cells with a synthetic phenotype (vacuolar degeneration, fissed mitochondria, and irregular cell polarization).4.Erythrocyte extravasation (due to rupture of the vasa vasorum) and neovascularization foci on the media-adventitia border.5.Electron microscopy signs of mitochondrial cytopathy (circular cristae and pathological inclusions in the matrix of mitochondria) and changes in the structure of extracellular elastic and collagen fibers.

The synthetic phenotype of medial smooth-muscle cells with vacuolar degeneration was also described for aortic dissections and aneurysms, atherosclerosis, and coronary artery dissections [16,17,18,19].

Several genome-wide association studies (GWASs) have identified CeAD-associated variants in loci-containing genes that encode regulators of the cytoskeleton (*PHACTR1* and *CCDC102B*) and mediators of the NO/VEGF/PDGF signaling pathways (*PLCE1*, *IRAG1*, and *LRP1*), ubiquitin-dependent protein degradation activator (*LNX1*), and transcription factors (*ZNF804A*, *FHL5*, and *FALEC*) [20,21]. Interestingly, these genes have valuable polymorphisms associated with other vasculopathies (migraine, coronary artery disease, arterial hypertension, and abdominal aortic aneurysm). Several polymorphisms with contradictory significance have been identified in genes encoding intercellular adhesion molecule 1 (*ICAM-1*), collagen type III (*COL3A1*), and methylenetetrahydrofolate reductase (*MTHFR*) [22]. The described polymorphisms lie predominantly within non-coding regions and may impact gene transcription and RNA processing [23]. However, global gene expression profile analysis has not yet been performed for CeAD patients.

Bulk RNA sequencing (RNA-Seq) is widely used for primary gene expression profiling in clinically available samples, such as peripheral blood or tissue biopsies. Gene expression profiling of peripheral blood using bulk RNA sequencing has already been performed for atherosclerosis [24] and aortic aneurysms [25,26].

To investigate the expression profile of protein-coding genes in the peripheral blood of CeAD patients, we conducted bulk RNA sequencing and differential gene expression analysis for 19 CeAD patients and 18 healthy volunteers. This study is promising in terms of identifying molecular markers of CeAD for patients with non-specific clinical signs, recurrent and complicated dissections, as well as identifying the relationships between CeAD, UCTD, and other vasculopathies (aortic and coronary artery dissection, aortic and cerebral aneurysms, atherosclerosis, tortuosity of arteries, pulmonary arterial hypertension, etc.).

## 2. Results

### 2.1. Differential Gene Expression Analysis with DAVID Functional Annotation

We identified 180 upregulated genes and 309 downregulated genes in the CeAD patients compared to healthy volunteers (adj. *p*-value < 0.05; Figure 1a, Appendix A). A substantial proportion (62%) demonstrated a significant change in expression (more than 2-fold, |log2 FC| > 1.0).

Functional annotation via DAVID revealed that the downregulated genes were mostly involved in translation, rRNA processing, Slit–Robo signaling, oxidative phosphorylation, and cell cycle regulation (Figure 1b, Appendix A). Moreover, the top downregulated genes included several immediate early transcription factors related to angiogenesis and vasculopathy, such as *JUN* (log2FC −1.76, adj. *p*-value 0.00068), FOSL1 (log2FC −3.89, adj. *p*-value 0.0069), and *EGR1* (log2FC −3.6, adj. *p*-value 0.021), and regulators of ribosomal metabolism and the MDM2-p53 pathway, such as *ANG* (log2FC −2.33, adj. *p*-value 0.01), *UBA52* (log2FC −1.33, adj. *p*-value 0.031), and *RPS27a* (log2FC −1.48, adj. *p*-value 0.013). The upregulated genes are predominantly responsible for stress-related signal transduction, cell migration and morphogenesis, and changes in phospholipid metabolism and cation transport (Figure 1c, Appendix A). Among the top upregulated genes were p53-dependent inducers of cell cycle arrest and apoptosis, such as *ACSM3* (log2FC 3.5, adj. *p*-value 0.00068), *SMPD3* (log2FC 4.0, adj. *p*-value 0.005), and *DACH1* (log2FC 3.11, adj. *p*-value 0.00008).

The analysis of cell-type-specific gene expression signatures using DAVID against the Human Protein Atlas (Tissue Cell Type section) demonstrated that 53–60% of the differentially expressed genes (DEGs) are highly specifically expressed in glandular epithelial cells (enrichment score 14.59, FDR < 1.1 × 10^−11^; Appendix A).

### 2.2. Gene–Disease Associations

We explored whether DEGs were involved in any previously described gene–disease associations (GDAs) using DisGeNET (Appendix A). We found that the largest proportion of DEGs were associated with neoplasms, mainly adenocarcinomas and leukemias (36.2% of DEGs; 19.2% with a GDA score > 0.1), neurological disorders (26.4% of DEGs; 14.1% with a GDA score > 0.1), and cardiovascular diseases (20.7% of DEGs; 9.4% with a GDA score > 0.1). A more detailed analysis of the associations with cardiovascular diseases revealed that 11% of the DEGs were associated with arterial hypertension, 10% with myocardial infarction, 9% with ischemic stroke, 8% with heart development anomalies, 7% with atherosclerosis, and 7% with coronary artery disease. We also observed that some DEGs were simultaneously associated with heritable connective tissue dysplasia, other vasculopathies, and their cerebral complications (Figure 2). The genes commonly shared between the diseases included some of the top dysregulated genes for CeAD, with more than three-fold change, such as *ACSM3* (log2FC 3.48, adj. *p*-value 0.00068), *EGR1* (log2FC −3.56, adj. *p*-value 0.021), *ANG* (log2FC −2.33, adj. *p*-value 0.01), *JUN* (log2FC −1.76, adj. *p*-value 0.00068), and *ABO* (log2FC 3.35, adj. *p*-value 0.005). Interestingly, the most commonly shared gene was *SOD1* (log2FC −1.08, adj. *p*-value 0.019), indicating that this outer-membrane mitochondrial protein is a potential common pathogenic factor for cerebrovascular pathologies.

We screened the reported genetic variants from GWASs of CeAD and migraine against the DEGs discovered in this study. Integrating a small-scale CeAD GWAS revealed no direct overlaps. The gene *PHACTR1*, reported as a top hit in this GWAS, was not significantly dysregulated in our data. However, the focal intronic genetic variant rs9349379 is located within 500 kb of the upregulated *GFOD1* (log2FC 2.68, adj. *p*-value 0.00008), suggesting a potential cis-regulatory mechanism influencing the molecular phenotype of CeAD. A large-scale migraine GWAS of more than 800,000 individuals [27] included multiple significant genetic variants located in close proximity (<50 kb) to 15 DEGs (Appendix A). These included the gene *SLC24A3* (log2FC 2.75, adj. *p*-value 0.00033), which directly overlaps with the migraine-associated genetic variants and is one of the top DEGs. *SLC24A3* encodes a sodium/calcium exchanger and is an important component of calcium homeostasis.

### 2.3. Heatmap Clustering

An analysis of the normalized expression levels of the 489 DEGs (adj. *p*-value < 0.05) revealed that the cohort could be divided into three groups (Figure 3):1.Healthy volunteers,2.Patients with the most pronounced stigmas of UCTD (>9 points),3.Patients, including all the most severe cases of CeAD (recurrent dissection, ICA/VA aneurysm, and combined ICA/VA dissection).

Additionally, we divided the DEGs on the heatmap into four clusters, reflecting the phenotypic status of the samples (Figure 3) and the gene functional properties (Figure 4, Appendix A):The first and second clusters (63 and 246 genes, respectively) contained genes with downregulated expression in both CeAD subgroups that were enriched in cell cycle regulation, translation, mitochondrial respiration, and ribosome processing (FDR < 0.05).The third cluster comprised 87 genes that were upregulated in both CeAD subgroups and enriched in calcium metabolism, G-protein-associated signaling, and the regulation of cell polarization and migration (FDR < 0.05).In the fourth cluster, 91 genes were only upregulated in the subgroup of patients with the most pronounced stigmas of UCTD and in some healthy volunteers. These genes were related to phagocytosis, the exchange of membrane lipids, and VEGFA-VEGFR2 signaling (FDR < 0.05).

We found that the most affected pathways across all four gene clusters were involved in translation, Slit–Robo signaling, rRNA processing, and oxidative phosphorylation in mitochondria, and over 15% of the genes in these pathways were downregulated in CeAD patients (Figure 4).

### 2.4. Protein–Protein Interactions

We used the STRING database to construct a PPI network of the DEGs (Figure 5). The network comprised several protein clusters. A large cluster of underexpressed cytosolic ribosomal proteins was connected to smaller clusters of overexpressed regulators of cell differentiation and migration, heterogeneous underexpressed mitochondrial proteins (I and V complexes of the respiratory chain, and transport and ribosomal proteins), and underexpressed regulators of rRNA processing, splicing, cell cycle, and apoptosis. Most of the smaller clusters were associated with putative extra-ribosomal functions of ribosome proteins (Appendix A).

Furthermore, the PPI network analysis revealed deregulated proteins with multiple interactions across several clusters simultaneously, indicating potential mediatory roles. In particular, we identified the top downregulated genes *UBA52* and *RPS27a* as putative mediators of ribosomal proteins and as (1) regulators of transcription (through the downregulated genes *JUN* and *ATF4*), (2) regulators of cell differentiation and migration (through the upregulated gene *CDC42*), and (3) regulators of cell cycle and apoptosis (partially through the downregulated nuclear genes *NPM1*, *POLR2L*, and *POLR2K*). The nucleolar protein *SNU13* connects cytoplasmic ribosomes with spliceosomes, while the *HSPA8* chaperone connects cytoplasmic ribosomes with markers of proteostasis and inflammation. The downregulated *TOMM7* and *SOD1* genes encode proteins of the outer mitochondrial membrane and link nuclear and mitochondrial resident proteins.

### 2.5. Gene Set Enrichment Analysis

Rank-based enrichment analysis of the overall 15,363 protein-coding genes (Figure 6, Appendix A) supported the findings on ribosomal and mitochondrial dysregulation in CeAD and provided further details on the specific pathways involved in cell cycle arrest and cell differentiation. We observed activation of the Wnt, MAPK, AMPK, Ras, and JAK-STAT pathways, which are involved in the regulation of cell differentiation, adhesion, and migration, as well as the upregulation of stress-related processes, such as phospholipid metabolism and endocytosis, calcium transport, and actin dynamic polymerization. In line with the analysis of DisGeNET, we observed that the transcriptomic profiles of patients with CeAD showed a downregulation of genes associated with neurodegenerative diseases (Alzheimer’s, Parkinson’s, Huntington’s, and prion diseases, and amyotrophic lateral sclerosis) and dysmetabolic diseases (non-alcoholic fatty liver disease and diabetic cardiomyopathy). We also observed an upregulation of genes associated with cardiovascular processes (e.g., hypertrophic cardiomyopathy, blood coagulation, and platelet activation) and neoplasms (e.g., pancreatic adenocarcinoma and chronic myeloid leukemia).

## 3. Discussion

In this study, we investigated the global expression profiles of protein-coding genes through RNA sequencing of peripheral blood cells from CeAD patients. Initially, we observed a concordant downregulation of genes involved in translation and rRNA processing. Mutations in this gene group have been linked to monogenic ribosomopathies [28,29], characterized by a diverse early-age phenotype (anemia, bone dysplasia, craniofacial dysmorphia, muscle atrophy, hearing loss, pancreatic insufficiency, dyskeratosis, hair hypoplasia, asplenia, etc.) and an increased susceptibility to cancer later in life. The causes of tissue-specific manifestations in systemic ribosomopathies are not fully understood. There is mounting evidence of distinct extra-ribosomal functions for many ribosomal proteins [30,31]. Consequently, significant changes in ribosomal protein levels are closely associated with alterations in cell phenotype and adaptive capacity under stress conditions [32,33]. Elevated expression of ribosomal proteins is linked to tumorigenesis and embryogenesis [34,35], while reduced expression supports mechanisms of cellular aging, differentiation, and integrated stress responses [36,37].

The synthesis of rRNA constitutes approximately 60% of cell transcriptional activity and occurs within the nucleoli. As a result, the nucleoli are highly susceptible to various stressors, such as nutritional deficiencies, viral load, and reactive oxygen species. The changes associated with disordered ribosome biosynthesis have been identified as nucleolar stress [38], which can arise from faulty rRNA processing and ribosomal assembly, DNA damage, viral overload, or nucleotide depletion. Nucleolar stress is characterized by a loss of nucleolar integrity and the relocation of resident nucleolar proteins, such as NPM1 (nucleophosmin), to the nuclear periphery [32,33]. These nucleolar proteins inhibit the MDM2 ubiquitin ligase, leading to the activation of the tumor suppressor p53. In turn, this halts cell proliferation, induces cell cycle arrest, cell differentiation, and cell aging, and triggers competitive mechanisms of apoptosis and autophagy [32,33,39]. The regulators of the MDM2-p53 pathway were identified as the top DEGs for CeAD in our study.

The significant role of nucleolar stress has already been shown for other vasculopathies. An inhibition of translation with a predominantly secretory phenotype in medial smooth myocytes has been observed in abdominal aortic aneurysm. It was suggested that defective assembly of the RNA polymerase I complex triggers nucleolar stress in this condition [40]. Nucleolar stress has been implicated in the pathogenesis of neurodegenerative diseases, such as amyotrophic lateral sclerosis and frontotemporal dementia [41], Parkinson’s [42], Alzheimer’s [43], and Huntington’s diseases [44], and spinocerebellar ataxia type 3 [45]. Our FGSEA also revealed an association between the gene expression profile for CeAD and these diseases.

Our study found that the DEGs in CeAD are primarily associated with neoplasms. Nucleolar stress counteracts tumorigenesis by limiting the pool of regulatory proteins for cell cycle promotion and DNA repair [32,33,39]. The connection between ribosome biogenesis and tumor transformation was supported by accelerated ribosome biogenesis in actively dividing tumor cells [46]. Spontaneous dissection of the coronary [47,48,49,50] and cervical arteries [51,52,53] has been described as a side effect of chemotherapy, which inhibits rRNA transcription (dactinomycin, doxorubicin, oxaliplatin, etc.) or VEGF signaling (bevacizumab, ranibizumab, sorafenib, etc.) [54,55].

As mentioned earlier, the level of ribosomal proteins depends on the cell phenotype. A cell with irreversible cell cycle arrest due to persistent damage is known as having the senescence-associated secretory phenotype (SASP) [56]. The activation of SASP is an effective mechanism for suppressing cancer, which is characterized by p53-dependent cell cycle arrest, the active secretion of proinflammatory cytokines, chemokines, and growth factors, the secretion of proteases that remodel the extracellular matrix, and changes in the dynamics of the actin cytoskeleton and cell adhesion [57]. The SASP phenotype plays a dual role in the development of neoplasms and inflammatory diseases. With prolonged exposure to a harmful factor, the number of cells with SASP increases—the oncosuppressive effects transform into oncogenic effects, stimulating metastasis, angiogenesis, and immunosuppression [56]. This could explain the development of cancer in older-age patients with ribosomopathies.

The FGSEA in this study identified the activation of cell adhesion, the restructuring of the actin cytoskeleton, and the co-activation of pathways (Wnt, Ras, MAPK, AMPK, JAK-STAT, IL-1, and IL-6 signaling) as driving oncogene-induced senescence in patients with CeAD. These changes correspond to the pathomorphological changes in vascular dissection (the synthetic phenotype of smooth myocytes with vacuolar degeneration), which were also described for an abdominal aortic aneurysm and arterial atherosclerosis [40].

Interestingly, the chronic SASP phenotype alters the extracellular microenvironment and facilitates the transition between epithelial and mesenchymal phenotypic features [58,59]. This process, known as epithelial–mesenchymal plasticity (EMP), is involved in organogenesis, tumorigenesis, and tissue regeneration with fibrosis [60,61]. The role of EMP has been previously described in atherosclerosis with unstable plaques, pulmonary artery hypertension, and cerebral cavernous malformations [62]. Our results indicated that the DEGs in mesenchymal cells of patients with CeAD are highly specifically expressed in glandular epithelial cells and associated with adenocarcinomas. We also observed a significant decrease in the expression of several regulators of epithelial–mesenchymal transition (*SNAI1*, *ATF4*, *UBE2S*, and *FOSL1*) for CeAD, along with an opposite activation of the pathways associated with cell migration and focal adhesion. Dysregulated EMP potentially explains the pathomorphological changes in CeAD (heterogeneous structure of the vascular wall with a loss of endothelial integrity and irregular polarization of smooth-muscle cells).

According to the FGSEA, patients with CeAD also had decreased expression of the mitochondrial genes encoding proteins of the respiratory chain, mitochondrial ribosomes, transport complexes through the inner and outer membranes, and cristae stabilizers. Previous electron microscopic studies of vessel walls after dissection indicated mitochondrial cytopathy (circular cristae and pathological inclusions in the mitochondrial matrix) [12,13]. Mitochondrial sub-activity is known to accompany cellular senescence [63]. Suboptimal levels of mitochondrial activity prevent the formation of reactive oxygen species, thereby reducing the cells’ requirements for DNA repair during the cellular integrated stress response [64]. Single-cell RNA sequencing revealed a downregulation of genes related to NADH complex assembly and ATP metabolic processes in formed and ruptured cerebral aneurysms in mice (especially in clusters of smooth-muscle cells with senescent secretory phenotype and activated focal adhesion) [65]. However, the hierarchy of interactions between nucleolar stress and decreased respiratory capacity in mitochondria is poorly understood. This crosstalk may involve nuclear–mitochondrial signaling, leading to mitochondrial fission, mitophagy, or AMPK-dependent downregulation of ribosome biosynthesis.

We identified two subgroups of patients with CeAD in the heatmap of the DEGs: the first subgroup had the most pronounced manifestations of UCTD (>9 points), and the second subgroup included all severe dissection cases (recurrent dissection, dissecting aneurysm, or combined ICA/VA dissection). Additionally, the gene clusters functionally annotated to ribosome biogenesis, SASP, and mitochondrial function showed similar expression in both subgroups, whereas the gene clusters functionally annotated to calcium metabolism, phagocytosis, and VEGF signaling were more active in the first subgroup. These findings suggest the potential heterogeneity of CeAD and the presence of compensatory mechanisms under nucleolar stress conditions, such as autophagy and mitophagy, the phagocytosis of apoptotic cells, and VEGF-mediated vessel wall regeneration. VEGF signaling, accompanied by calcium metabolism activation, stimulates the phagocytosis of apoptotic cells [39,66]. This explains the development of secondary vascular dissections following chemotherapy with VEGF signaling inhibitors. Depending on the ability of these compensatory mechanisms to limit the adverse effects of chronic SASP and nucleolar stress, dissection may primarily develop in individuals with UCTD. Alternatively, more severe dissection can develop in patients with fewer signs of UCTD.

The findings of this study must be considered within the context of certain limitations. The results of transcriptome analysis are highly dependent on sample collection and preparation conditions. Therefore, we included RNA isolation and library preparation batches in the design of bioinformatics processing. Expression profiling of protein-coding genes does not account for the subsequent influence of post-transcriptional modifications and non-coding RNAs on the profile of protein expression, which could differ significantly from the gene expression profile. Hence, this study aimed to find potential correlations between CeAD and several molecular pathways that could be precisely studied in the future using fluorescent microscopy and tandem mass spectrometry on biopsies from sporadic and chemotherapy-induced dissections. Bulk RNA sequencing techniques cannot assess the contribution of individual cell populations in nucleolar stress, SASP, epithelial–mesenchymal plasticity, and mitochondrial dysfunction processes. To clarify the data obtained, modern single-cell sequencing techniques should be used on blood and vessel wall biopsies. Finally, it is essential to increase the sample size in future studies to enable more detailed within-group comparisons of differential gene expression between patients with vertebral and internal carotid artery dissections, varying degrees of connective tissue dysplasia, ischemic and non-ischemic manifestations of dissection, and recurrent and single dissections in anamnesis.

## 4. Materials and Methods

### 4.1. Clinical Description of the Cohort

A total of 19 patients with VA and/or ICA dissection (13 women and 6 men; mean age 37.6 ± 3.95 years) were included in the study group. CeAD verification was performed by detecting intramural hematoma on MRI in T1 fat-saturated mode and/or characteristic angiographic signs of dissection within 5 days to 3 months after dissection. The patients were observed at the Research Center of Neurology (Moscow, Russia) for at least 12 months after the last dissection prior to RNA sampling. The clinical characteristics of the patients in the study group are detailed in Table 1.

All the patients in the study group met at least one of the following criteria for possible UCTD:1.A dissection of two or more cervical arteries (10 cases).2.A recurrent dissection during 1 year of observation after the primary CeAD (3 cases).3.An intense chronic headache presenting long before the development of CeAD (13 cases).4.A dissecting aneurysm of the VA and/or ICA diagnosed with MR angiography (4 cases).5.The presence of more than 8 stigmas of connective tissue dysplasia selected from criteria of known monogenic heritable disorders of connective tissue (13 cases; Appendix A) [67].

The control group consisted of 18 healthy volunteers (12 women and 6 men; mean age 30.1 ± 6.65 years). Healthy volunteers were selected from the Research Center of Neurology staff, who underwent continuous clinical observation. None of them had chronic headache, intense neck pain, transient ischemic attacks, or stroke in anamnesis. They did not exhibit signs of connective tissue dysplasia.

A detailed description of the study population is shown in Appendix A.

This study was approved by the Research Center of Neurology Ethics Review Board (Protocol No. 6-4/22, dated 6 June 2022). Each participant provided informed consent for participation and the processing of personal data. The exclusion criteria included refusal to participate in the study, severe somatic and infectious diseases, and traumatic VA and/or ICA dissection.

### 4.2. Sample Processing

Peripheral blood samples were collected in 4 mL EDTA vacutainer tubes and stored at 4 °C for up to 2 h. Total RNA was extracted from 1.5 mL of fresh blood samples using the RNeasy Mini Kit (Qiagen, Germantown, MD, USA). Total RNA samples were then purified from residual DNA contaminants using Monarch DNase I enzyme (New England BioLabs, Ipswich, MA, USA). The concentration of the RNA samples was measured using a Qubit fluorometer with the Qubit RNA HS Assay Kit (Invitrogen, Waltham, MA, USA). The RNA integrity number (RIN) was measured on an Agilent TapeStation 4150 system using the High-Sensitivity RNA ScreenTape (Agilent, Santa Clara, CA, USA) to indicate the quality of the total RNA samples. The RIN values for the samples ranged from 8.7 to 9.4.

Libraries for bulk RNA-seq were prepared from 1000 ng of total RNA per sample using the TruSeq Stranded Total RNA Library Prep Gold Kit (Illumina, San Diego, CA, USA), with prior removal of cytoplasmic and mitochondrial fractions of ribosomal RNA. The IDT for Illumina TruSeq RNA UD Indexes (Illumina, San Diego, CA, USA) were used for library indexing. Library quality assessment was performed on the Agilent TapeStation 4150 system using the High-Sensitivity D5000 ScreenTape and High-Sensitivity D5000 Reagents (Agilent, Santa Clara, CA, USA). The molar concentration of the libraries was measured via real-time PCR using the KAPA Library Quantification Kit (KAPA Biosystems, Wilmington, MA, USA). The obtained equimolar libraries were pooled and sequenced on the Illumina NovaSeq 6000 platform (Illumina, San Diego, CA, USA) in paired-end mode, with a length of 151 + 151 bp. There was an average of 38 million paired reads per sample.

### 4.3. Bioinformatics Processing

The sequencing reads were demultiplexed based on index sequences with bcl2fastq version 2.20.0.422. Alignment of the reads onto the annotated human genome (GRCh38) was performed with STAR version 2.7.10a [68], with an average alignment rate of 80%. Approximately 60% of the reads were uniquely mapped to annotated genes and summarized into counts using HTSeq version 2.0.2 in “union” mode [69]. Further analysis was performed for 20,013 protein-coding genes. A summary of the pre- and post-sequencing data for all the samples is shown in Appendix A.

EdgeR version 3.40.2 was used to perform trimmed mean of M-values (TMM) normalization and filter out 4650 genes with counts of less than 0.5 per million in half or more samples [70]. EdgeR was applied to estimate dispersion values using housekeeping genes, which have been reported to be the genes with the most stable expression in all human tissues (*C1orf43*, *CHMP2A*, *EMC7*, *GPI*, *PSMB2*, *PSMB4*, *RAB7A*, *REEP5*, *VCP*, and *VPS29*) [71]. Differential expression analysis of 15,363 protein-coding genes was performed using EdgeR with a multiple linear regression model, taking into account age, sex, and sample preparation batches as covariates (patients with CeAD vs. healthy volunteers). Gene expression changes were considered significant with any log2 fold change (log2FC) if the adjusted *p*-value was <0.05.

Volcano plots for the differentially expressed genes (DEGs) were generated with iDEP version 1.1 [72]. Functional annotation for the downregulated and upregulated genes was performed using DAVID version 6.8 [73] against the Reactome [74], KEGG [75], GO BP [76], and WikiPathways [77] databases (similarity threshold > 0.8; FDR < 0.05). The tissue-specific expression of the DEGs was analyzed using DAVID against the Human Protein Atlas (Tissue Cell Type section) [78], with a similarity threshold > 0.8 and an FDR < 0.05. Known gene–disease associations (GDAs) in humans were searched using the DisGeNET database [79]. The statistical significance of overlap between the GWAS and DEGs was estimated using the “regioneR” R package version 1.30.0 [80], with 10,000 permutations.

The heatmaps for the DEGs were generated with iDEP. Gene expression levels were represented as normalized logCPM (counts per million) values and transformed into a standard score (Z score), with prior correction using the “removeBatchEffect” function from the “limma” R package version 3.54.2. Pearson correlation estimation was used to cluster the samples on the heatmap. Functional annotation clustering for the DEGs on the heatmap was performed using the “compareCluster” function of the “clusterProfiler” R package version 4.6.2 [81] and iDEP based on the Reactome, KEGG, GO BP, and WikiPathways databases. The number of gene clusters on the heatmap was defined in such a way that every cluster had the largest number of annotated pathways with the smallest false discovery rate values (FDR < 0.05). The “dotplot” function of the “clusterProfiler” R package was used to visualize the annotation of the gene clusters.

Protein–protein interaction (PPI) analysis of the DEGs was performed using STRING version 12.0 [82]. PPIs with an interaction score of at least 0.5 were shown on the diagram. Overall, 15,363 protein-coding genes were ranked using “sign(log2FC) × −log10(*p*-value)” and involved in the enrichment analysis of molecular pathways using the “clusterProfiler” R package in the “FGSEA” mode against the KEGG, GO BP, and WikiPathways databases. Enrichment was considered significant if the q-value was <0.05. The “ridgeplot” function of the “clusterProfiler” R package was used to visualize the enrichment analysis results.

## 5. Conclusions

Bulk RNA sequencing data suggested the potential involvement of nucleolar stress, chronic senescence-associated secretory phenotype, mitochondrial malfunction, and dysregulated epithelial–mesenchymal plasticity in the pathogenesis of CeAD. Phagocytosis and VEGF-mediated angiogenesis may reduce the risk of severe artery dissection. These findings underscore the importance of further exploring the connections between cervical artery dissection and tumorigenesis. Non-coding gene expression profiling analysis is necessary to complete the obtained data. Histological studies of nucleolar stress and phenotype switching in biopsy specimens are essential, particularly in cases of chemotherapy-related vessel dissection. Prospective epidemiological studies investigating cancer incidence among CeAD patients could provide valuable insights.

## Figures and Tables

**Figure 1 ijms-25-05205-f001:**
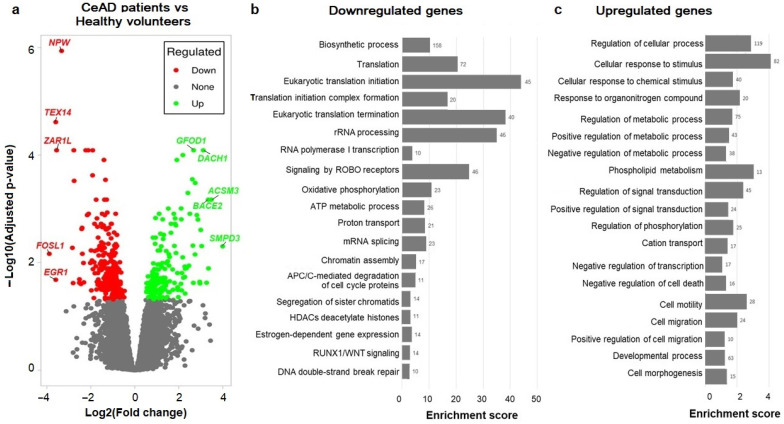
Differential expression analysis of protein-coding genes in patients with cervical artery dissection (CeAD) vs. healthy volunteers: (**a**) Volcano plot with 180 upregulated genes (green) and 309 downregulated genes (red). The top 10 genes with the largest |log2FC|are labeled. (**b**,**c**) Enrichment score plots for the top representative pathways enriched in downregulated and upregulated genes. The number of differentially expressed genes (DEGs) involved in the pathway is indicated near each line.

**Figure 2 ijms-25-05205-f002:**
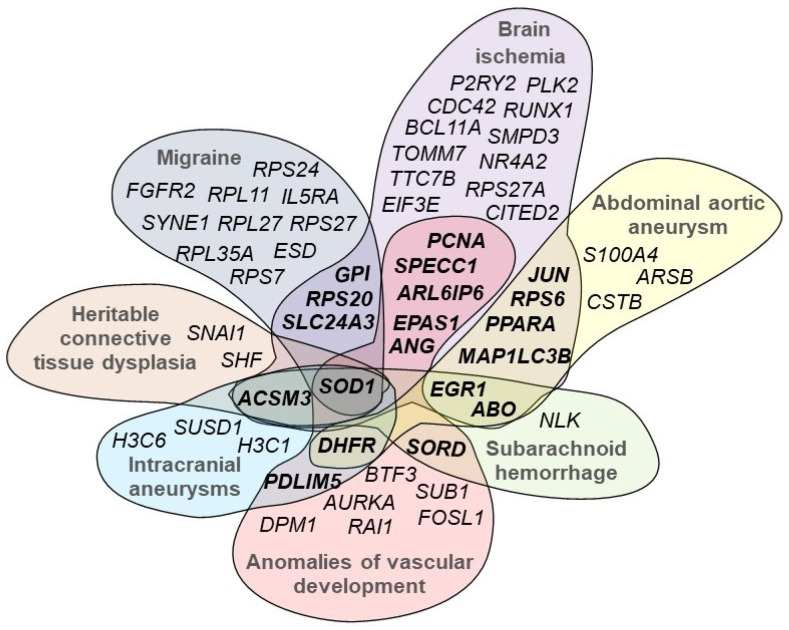
Cross-GDA analysis of CeAD DEGs between heritable connective tissue dysplasia, other vasculopathies, and their complications. The commonly shared genes are shown in bold.

**Figure 3 ijms-25-05205-f003:**
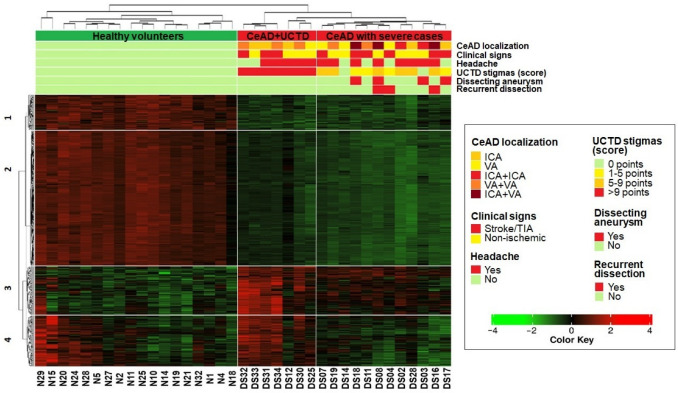
Heatmap of the 489 DEGs with normalized expression levels (transformed into a standard Z score). Three phenotypic groups are marked vertically. Four gene clusters are marked horizontally. UCTD—undifferentiated connective tissue dysplasia; ICA—internal carotid artery; VA—vertebral artery; TIA—transient ischemic attack.

**Figure 4 ijms-25-05205-f004:**
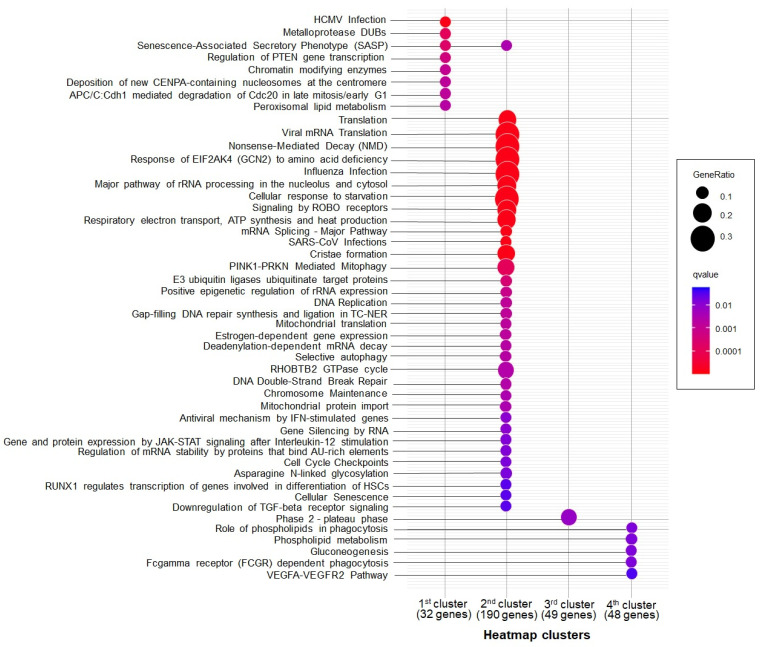
Functional annotation of the gene clusters from the heatmap generated with clusterProfiler against the Reactome database. The number of annotated genes for each cluster is labeled on the x-axis.

**Figure 5 ijms-25-05205-f005:**
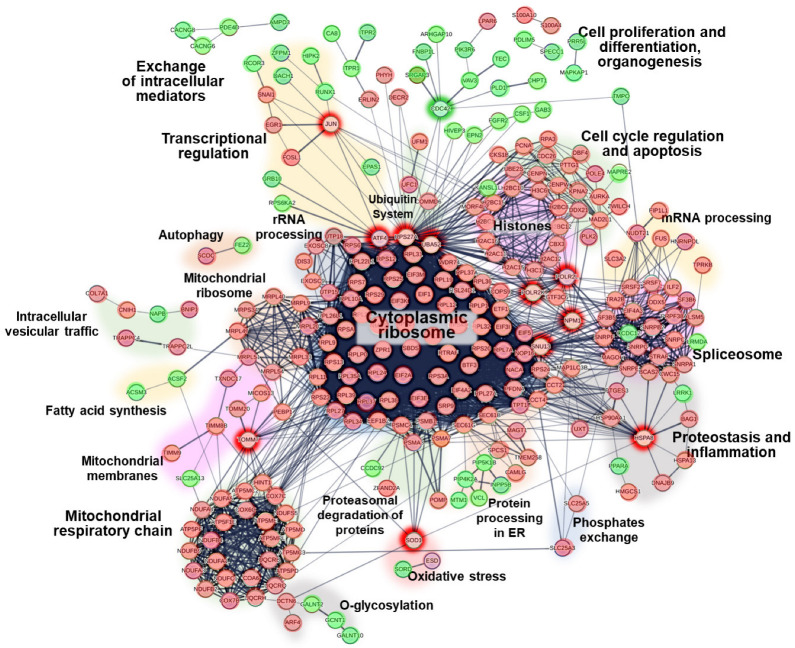
The protein–protein interaction (PPI) network of the DEGs for CeAD built with the STRING database. Proteins underexpressed in CeAD are shown in red, while overexpressed proteins are shown in green. The clusters of proteins with multiple strong internal PPIs are highlighted and named. The potential mediators between protein clusters are additionally highlighted.

**Figure 6 ijms-25-05205-f006:**
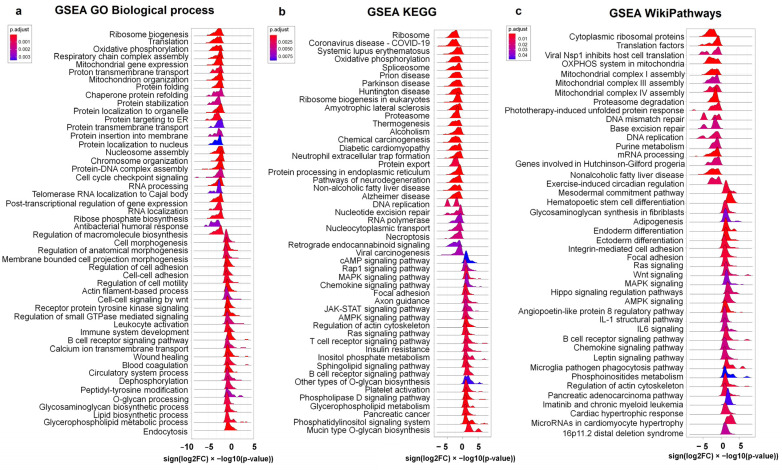
The ridge plots of GSEA results for CeAD against the GO BP (**a**), KEGG (**b**), and WikiPathways (**c**) databases. The distribution of rank orders for the individual genes is displayed on the x-axis. Similar pathways are located next to each other on the y-axis.

**Table 1 ijms-25-05205-t001:** Clinical characteristics of CeAD patients in the study group.

Characteristics	Number of Patients		Ischemic Stroke/Transient Ischemic Attack Due to Dissection		Isolated Headache/Neck Pain Due to Dissection		UCTD Stigmas	Chronic Headache in Anamnesis	
	N	%	N	%	N	%	Mean Points Score	N	%
All	19	100	9	47	10	53	8.9 ± 3.8	13	68
Women	13	68	4	21	9	48	10.9 ± 3.7	10	52
Men	6	32	5	26	1	5	4.2 ± 3.5	3	16
ICA dissection	5	26	2	22	3	30	9.6 ± 3.7	3	23
VA dissection	4	21	2	22	2	20	8.6 ± 4.7	2	15
Dissection of two or more cervical arteries	10	53	5	32	5	50	8.6 ± 3.5	8	62

## Data Availability

The original contributions presented in this study are included in the article/Appendix A. Further inquiries can be directed to the corresponding author.

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
