# Peer review of "Peripheral Blood Gene Expression Profiling Reveals Molecular Pathways Associated with Cervical Artery Dissection"

_ijms, 2024, doi:10.3390/ijms25105205_

Round 1

Reviewer 1 Report

Comments and Suggestions for Authors

Dear Authors,

The problem of cervical artery dissection and its genetic background. There are some major and minor issues related to methodology were detected and need to be adressed before accepting the manuscript.

1) What was the pattern of control group enrollment? Any matching involved or just random healthy volunteers?

2) Did you calculate power and sample size? Why 19 participants?

3) Did you observe the diferences in characteristic or obtained results between ICA and VA dissection patients? It is know that mechanisms of arterial dissections may have different backgraound depending on vessel type.

4) There is lack of more detailed study population description, additional table with the group characteristis.

5) Study limitations subsection should be added in Discussion section.

Kind regards

Comments on the Quality of English Language

Moderate editing of English language required

Reviewer 2 Report

Comments and Suggestions for Authors

1. The standard arrangement of subchapters is: 

Introduction=> Materials and Methods=> Results=>Discussion and Conclusions, not different. 

2. Apart from that, the manuscript seems to be written neatly and correctly, I have no major comments.

3. I cant see the limitations section. 

Comments on the Quality of English Language

No major corrections are needed

Author Response

Dear Reviewer,

thank you for the review of our work. We hope that our edits adequately address each of your comments below.

  1. The standard arrangement of subchapters is: Introduction=> Materials and Methods=> Results=>Discussion and Conclusions, not different. Dear

Dear Reviewer, we followed the subchapter arrangement provided on the journal's website and in the journal’s ready-made template. We also checked how subchapters are presented in other published articles within the journal.

  1. Apart from that, the manuscript seems to be written neatly and correctly, I have no major comments.

Dear Reviewer, we appreciate your positive evaluation of our manuscript's quality and value all the comments provided by you.

  1. I cant see the limitations section.

The findings of this study must be considered within the context of certain limitations. The results of transcriptome analysis are highly dependent on sample collection and preparation conditions. Therefore, we included RNA isolation and library preparation batches in the design of bioinformatics processing. Expression profiling of protein-coding genes does not account for the subsequent influence of post-transcriptional modifications and noncoding RNAs on the profile of protein expression, which could differ significantly from the gene expression profile. Hence, this study aimed to find potential correlations between CeAD and several molecular pathways that could be precisely studied in the future using fluorescent microscopy and tandem mass spectrometry on biopsies from sporadic and chemotherapy-induced dissections. Bulk RNA sequencing techniques cannot assess the contribution of individual cell populations in nucleolar stress, SASP, epithelial–mesenchymal plasticity, and mitochondrial dysfunction processes. To clarify the data obtained, modern sin-gle-cell sequencing techniques should be used on blood and vessel wall biopsies. Final-ly, it is essential to increase the sample size in future studies to enable more detailed within-group comparisons of differential gene expression between patients with vertebral and internal carotid artery dissections, varying degrees of connective tissue dysplasia, ischemic and nonischemic manifestations of dissection, and recurrent and single dissections in anamnesis (lines 359-376 in the corrected version of our manuscript).

Reviewer 3 Report

Comments and Suggestions for Authors

The authors conducted the comprehensive analysis of the gene expression in CeAD patients. In my opinion, such a study of gene expression profiling targeting patients contributes not only to understanding the overall picture of pathophysiology but also to elucidating the etiology. Since this paper provides valuable insights through diverse analytical perspectives, I therefore recommend that this paper acceptable in IJMS.

Author Response

Dear Reviewer, we appreciate your positive evaluation of our manuscript's quality and value all the comments provided by you.

Round 2

Reviewer 1 Report

Comments and Suggestions for Authors

Dear Authors,

Thank you for the improvements and explanations. At this point, I do not have any further questions or requests.

Congratulations on your work.

Kind regards

Comments on the Quality of English Language

Minor spelling corrections are suggested.